# Clinical Significance of Circulating Tumor Cells in Gastrointestinal Carcinomas

**DOI:** 10.3390/diagnostics10040192

**Published:** 2020-03-30

**Authors:** Leonie Konczalla, Anna Wöstemeier, Marius Kemper, Karl-Frederik Karstens, Jakob Izbicki, Matthias Reeh

**Affiliations:** Department of General, Visceral and Thoracic Surgery, University Medical Centre, Hamburg-Eppendorf, Martinistr. 52, 20246 Hamburg, Germany; l.konczalla@uke.de (L.K.); a.woestemeier@uke.de (A.W.); m.kemper@uke.de (M.K.); k.karstens@uke.de (K.-F.K.); izbicki@uke.de (J.I.)

**Keywords:** liquid biopsy, cancer diagnosis, solid cancer/tumor, circulating tumor cells, free DNA

## Abstract

The idea of a liquid biopsy to screen, surveil and treat cancer patients is an intensively discussed and highly awaited tool in the field of oncology. Despite intensive research in this field, the clinical application has not been implemented yet and further research has to be conducted. However, one component of the liquid biopsy is circulating tumor cells (CTCs) whose potential for clinical application is evaluated in the following. CTCs can shed from primary tumors to the peripheral blood at any time point during the progress of a malignant disease. Following, one single CTC can be the origin for distant metastasis at later cancer stage. Thus, CTCs have great potential to either be used in cancer diagnostics and patient stratification or to function as a target for new therapeutic approaches to stop tumor dissemination and metastasis at the very early beginning. Due to the biological fundamental role of CTCs in tumor progression, here, we provide an overview of CTCs in gastrointestinal cancers and their potential use in the clinical setting. In particular, we discuss the usage of CTC for screening and stratifying patients’ risk. Moreover, we will discuss the potential role of CTCs for treatment specification and treatment monitoring.

## 1. Introduction

Cancer is one of the leading causes of death worldwide, accounting for approximately 9.6 million deaths in 2018 and 17 million predicted incidences of newly diagnosed cases. Within these, around one third of cancer related deaths accounts to solid gastrointestinal cancers, including esophageal, gastric, pancreatic and colorectal cancer [1]. Precisely, patients do not die because of their primary tumor. It is the metastatic disease, which compromises organ function of, i.e., liver or lung. Therefore, it is crucial to prevent metastasis or treat metastatic lesions at early stage. However, radiological and endoscopic imaging techniques have low sensitivity rates to detect micrometastatic status. Thus, additional diagnostic tools are urgently needed. In recent years, disseminated tumor cells (DTC) in the bone marrow and circulating tumor cells (CTC) in peripheral blood were intensively evaluated as prognostic markers. Their relevance has been investigated in several studies and overall, their presence is associated with poor prognosis and high probability of metastatic disease [2]. However, CTC biology provide much more information than simple enumeration: precise detection and characterization of CTCs can pave the way for a new and easily accessible tool for diagnostic, patients’ outcome prediction, stratification of treatment and therapy monitoring (Figure 1). Thus, the clinical utility of CTC detection and analysis holds great promise. Here, we summarize the current knowledge about the potential and pitfalls of CTC biology and detection as well as their implications of clinical usage.

Already at early stage of tumor growth, the metastatic cascade starts with single circulating tumor cells (CTCs), which are shed into the bloodstream [3,4,5]. However, only the minority of CTCs are able to become a solid metastatic lesion due to many steps of the metastatic cascade. This includes the migration from the primary tumor through the endothelium into the blood, the survival and proper immune escape of the CTC into the blood stream and finally also the extravasation to distant organs and metastatic progress. Of note, in gastrointestinal cancers the numbers of CTCs in peripheral blood are already diminished in comparison to, e.g., breast cancer due to the portal vein circulation and a steady ‘first-pass-effect’ in the liver [6]. However, these complex mechanisms underscore that viable CTCs have a special phenotype. Thus, not only is the number of CTCs relevant, the molecular profile of CTCs influences their clinical significance as well.

In the following section, we provide an overview about the potential of CTCs in gastrointestinal cancers and their clinical application (Table 1). In particular, we discuss the significance of CTC detection in early cancer diagnosis and prognosis prediction. Moreover, we review the clinical application of CTCs in therapy selection and monitoring of cancer progress by sequential blood analyses.

## 2. Methods

A comprehensive search of electronic databases (PubMed, ScienceDirect, and Google scholar) using the key words “circulating tumor cells and/or liquid biopsy and/or staging and gastrointestinal cancer and/or solid cancer” and “prognosis or therapy or clinical monitoring or molecular marker” was conducted. The reference lists provided by the identified articles were additionally hand-searched for additional studies missed by the search strategy, and this method of cross-referencing was continued until no further relevant publications were identified. Evidence from these data was critically analyzed and summarized to produce this article with a broad overview about the most relevant gastrointestinal tumors and CTCs.

## 3. CTC Enrichment and Detection Methods

The identification and characterization of CTCs require highly sensitive methods since they are by far the minority of cell types in the peripheral blood. In the following section, we provide a brief update on CTC enrichment and detection methods to achieve a highly sensitive read out of liquid biopsy. Analysis methods of peripheral circulating tumor DNA (ctDNA), which can be done in parallel, have been reviewed elsewhere [29,30].

At the first step, cellular differences of cancer cells and other blood components are used to enrich CTCs from a blood sample. These can be either physical properties as size, densities or electrical charges of cancer cells and non-malignant blood cells or the differential expression of cell surface proteins on the cell surface [31,32]. In marker dependent techniques, mostly the epithelial cell adhesion molecule (EPCAM) is used as a cell surface protein for positive CTC selection [33]. Moreover, specific surface antigens (e.g., EGFR) and tissue-specific antigens (such as HER2) are used for positive selection enrichment methods. However, CTCs are known to be a heterogenous cell population with inconsistent expression patterns [34]. To avoid this pitfall, negative selection-based enrichment methods can be used instead: non-malignant blood cells are depleted after antigen-antibody- binding of characteristic surface markers, such as CD45 for leukocytes, CD146 for endothelial cells and CD34 for hematopoietic stem cells [35]. In contrast to positive selection, this strategy might reveal in lower purity of the isolated CTCs.

Hereinafter, CTC detection methods are used to identify real CTCs in the cell populations after enrichment, which may still contain thousands of non-malignant cells. Detection can be fulfilled by various approaches: to enable CTC quantification nucleic acid-based strategies are the gold standard since they allow to detect also mRNA or DNA of low abundance [36]. Moreover, functional assays, such as EPISPOT and EPIDROP, have been introduced to detect viable CTCs in short-term cell cultures of blood or bone marrow samples to enumerate viable CTCs and furthermore, characterize their secreted proteins [37,38]. Yet, the predominant detection approach uses, in parallel to enrichment methods, antibodies for immunological labeling of epithelial, mesenchymal or tissue-specific and tumor-associated markers [33]. In the following step, the fluorescence labeled antibodies can be detected through fluorescence microscopy, as in the US Food and Drug Administration (FDA) -approved CellSearch® system, or through fluorescence-activated cell sorting, if the cell enrichment results in a high CTC concentration [39,40,41] (Figure 2).

## 4. CTCs as Screening Tool

Frequently, gastrointestinal cancers, as esophageal, gastric, pancreatic or colorectal cancer, tend to be diagnosed at advanced tumor stage due to late onset of symptoms and the lack of highly sensitive imagine techniques to detect minimal tumor disease stages. Moreover, most screening methods are invasive and therefore, only conducted in patients at risk, e.g., with diagnosed precancerous lesions as a Barrett esophagus. Finally, radiological imaging as well as suspicious lesions during endoscopy needs further validation by tissue biopsy and pathological assessment. However, there are several pitfalls like a potential harmful invasive examination on the one hand, and sampling errors due to intratumoral heterogeneity on the other hand [42]. Contrarily, using peripheral blood is an elegant, easily accessible medium for patient screening. This is also one reason why in the past, research has been focused on possible tumor marker in the blood, e.g., the carcinoembryonic antigen (CEA). However, to date, there have been no specialized tumor markers which can be used as a specific tool to diagnose cancer [43].

Instead, the usage of CTCs and circulating tumor DNA (ctDNA) has been proposed as an easy screening tool to facilitate an early diagnosis of malignant diseases with high specificity and sensitivity. The CellSearch® system is the only CTC enrichment tool, which is approved by the FDA to detect CTCs. The CellSearch® system can be used to screen patients’ peripheral blood for CTCs to predict malignant disease [39]. Remarkably, a study in gastric cancer indicated that the presence of CTCs can distinguish between cancer patients and healthy controls with a sensitivity of 85.3% and specificity of 90.3% [7]. In parallel, a small study with squamous cell carcinoma patients of the esophagus revealed similar results which showed the ability of CTCs to differentiate between patients with and without malignant disease with high sensitivity and specificity [44]. This opens a new field of non-endoscopic and less-invasive screening for malignancies. However, this has to be taken with a grain of salt, since this cannot be assigned to all tumor entities. For example, screening for pancreatic ductal adenocarcinoma (PDAC) by liquid biopsy has low specificity due to the fact that other non-malignant diseases of the pancreas like chronic pancreatitis can lead to high rates of false positive epithelial cell numbers in the blood [8,9].

In that area of research, additional new and profound insights are sought by the prospective ICELLATE2 study of Castro et al. [26]. The ICELLATE2 study recruited 3388 subjectively healthy individuals with high risk factors for cancer. All of them were screened for CTCs and, if positive tested, enrolled in a special surveillance program for early detection of any cancer entity. This study might disclose promising results for early-stage-disease detection methods.

In addition to CTCs, ctDNA has been found in CTC-negative patients. Hence, ctDNA is discussed as a potential screening marker as well, even if the sensitivity seems lower in comparison to CTC detection [44,45]. In contrast, the CancerSEEK platform claims to detect eight different cancer entities with a sensitivity of 70% by ctDNA analysis of peripheral blood [46]. However, multicenter studies are still missing so that cancer prediction by CTCs or ctDNA can be implemented into clinical use as a reliable screening tool and liquid biopsy.

## 5. Cancer Staging and Patients’ Prognosis

Several studies in gastrointestinal cancers showed a positive correlation in between the presence of CTCs at timepoint of diagnosis and local invasion, malignant spread to lymph nodes and distant organs, as well as negative correlation with the progression-free survival and overall survival [12,13,47]. Prospective studies revealed that also patients with positive CTC score but without detectable lymph node or distant metastasis were at high risk for tumor recurrence and had a shorter overall survival [12,14,48,49].

CTC detection in solid cancers can improve the accuracy of prognosis and extend the risk stratification for patients with occult micrometastases. In the future, patients at risk could be selected for either close surveillance or for enhanced multimodal therapy by using CTC detection techniques. For example, Reeh et al. proposed to use CTC detection for preoperative patient staging in esophageal cancer. Thus, patients who are prone for tumor recurrence could be selected and treated intensively for the best possible outcome [15]. In parallel, another study revealed the potential of CTCs to predict metastatic spread and overall survival in pancreatic cancer [36].

Even in non-metastatic patients, CTCs can help to predict patients’ prognosis, for example, in esophageal cancer and PDAC [14,17]. Since CTC counts in non-metastatic patients are low, Buscail et al. proposed to improve the prognostic significance by comparing CTC counts with detection and gene expression measurements of tumor exosome DNA, revealing a highly sensitive and specific tool to predict PFS and OS [17]. However, it has to be mentioned that blood samples have partially been taken from the portal vein during operation. Therefore, higher numbers of CTCs and tumor DNA can be found into the blood in comparison to a normal peripherally blood samples after liver passage.

Amantini et al. reported that the CTC count as well as the expression profile of CTCs help to predict prognosis: palliative PDAC patients with CTCs characterized by high expression of ALCAM, POU5F1B and SMO have shorter PFS and OS compared to patients with CTCs with low mRNA levels of the depicted proteins [18].

In solid cancers, CTCs reveal to be a biomarker with great potential to improve the staging accuracy of cancer patients and provide a rational for detecting patients at risk and prediction of prognosis. Thus, the clinical implementation of CTC detection could help to stratify cancer patients and adjust their multimodal therapy.

## 6. Therapy Alignment

With new staging properties, the therapeutic regimes of cancer patients could be adjusted to patients’ risk and CTC count could help to provide the best individual therapy to every patient. For example, Tie et al. proposed to accelerate neoadjuvant and adjuvant chemotherapy regimens in patients with locally controlled colorectal cancer and positive CTC detection [50]. Accordingly, in esophageal and gastric cancer similar approaches have been proposed aiming to use CTC detection as prognostic value of therapy efficacy. However, not only the efficacy of chemotherapeutic therapy could be evaluated by CTC status, also radiotherapy efficacy could be estimated by pre and post radiation CTC count [19]. However, for final implementation of CTCs as a prognostic tool for locally controlled cancer lesions, further clinical studies are needed [15].

Further, the genetic information of CTC can help to adjust chemotherapy regimens. A mutation analysis in CTCs can open new insights in the efficacy of chemotherapeutic drugs. Brabender et.al. showed that in esophageal cancer, ERCC1 expression is associated with better response to neoadjuvant radio chemotherapy [20]. Additionally, evidence has been given that mutational analysis of target genes like KRAS or BRAF helps to predict the efficacy of targeted therapies like EGFR-monoclonal antibodies [21,22,51].

However, a study with 34 colorectal cancer patients showed that the mutations of primary tumor, CTCs and also ctDNA are not matching in every patient and the genetic heterogeneity in between the different tumor compartments is high [23]. These data underscore the clinical relevance of CTCs to treat all cancer cells of a patient with the best possible treatment regime and the often-noticed discordance between the genomic profile of the primary tumor and CTCs should not be underestimated [24]. A study of 88 metastatic gastric cancer (GC) patients demonstrated different HER2 status of primary tumor and CTCs, which enables new approaches for targeted therapies and clinically significant treatment reassignment [21,22,52]. These studies indicate that molecular analysis of CTCs can give valuable insights to choose the best possible treatment. In the future, liquid biopsy might be a sufficient substitute for invasive biopsy of primary tumors.

## 7. Longitudinal Therapy Monitoring

Therapy monitoring and early detection of cancer recurrence is an important topic in the field of oncology. In particular, non-invasive tests are favored to allow sequential screening. In the past, the value of CTCs in the field of therapy monitoring has been investigated.

In gastric cancer, it has been shown that patients can be clustered into chemotherapy responders and non-responders on the basis of the reduction of CTC numbers pre- and post-chemotherapy application [25]. However, even with diminished CTCs after chemotherapy, the study did not reveal a clear benefit in the overall survival of the patients. In contrast, in colorectal cancer reduced or negative CTC detection after chemotherapy appeared to be a significant value for survival prediction and measurement of chemotherapy efficacy [26]. Additionally, Lu et al. also showed that CTC screening after curative resection of colorectal cancers is a superior predictive marker for tumor relapse in comparison to the clinically implemented measurement of carcinoembryonic antigen (CEA) [27].

In esophageal cancer, changes in CTC detection after radiotherapy was found to be a prognostic value to predict progression free survival, irrespectively of chemotherapeutic agents [19]. A first pilot study, ESO-CTC, is running in the German cancer centers of Freiburg and Hamburg-Eppendorf which evaluates CTC counts during the phase III ESOPEC trial to compare perioperative and neoadjuvant chemoradiation in esophageal cancer. This prospective study will also shed light to the potential of CTCs in case of recurrence evaluation and therapy monitoring [53]. However, to assess the full potential of CTCs as a diagnostic tool for therapy efficacy and recurrence monitoring, more clinical studies are needed to define the CTC’s potential as biomarker in gastrointestinal cancers.

## 8. Conclusions

CTCs display a promising field to improve oncological treatment in gastrointestinal cancers. Quantitative enumeration of CTCs in peripheral blood has the ability to function as prognostic and treatment stratification for patients at risk. Detection of minimal residual disease enables to define patients at risk for cancer recurrence and distant metastasis. CTC detection could improve staging sensitivity and treatment specificity. Moreover, sequential blood analyses could lead to advanced sensitivity during surveillance monitoring of cancer patients. Molecular characterization of CTCs rises new possibilities of treatment alignment. Tumor-specific mutations as *KRAS* could be detected by the usage of CTCs and could steer the treatment to the best possible multimodal regime for every single patient. Finally, CTC detection is a promising value to assess the therapy efficacy in gastrointestinal tumors. However, prospective multicenter studies are still missing to validate CTCs in different gastrointestinal tumor entities for daily clinical application.

## Figures and Tables

**Figure 1 diagnostics-10-00192-f001:**
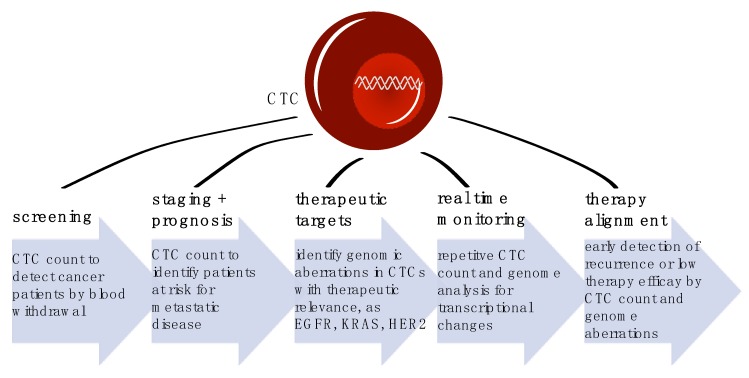
Clinical applications of CTC diagnostic in cancer patients at different timepoints of disease and therapy.

**Figure 2 diagnostics-10-00192-f002:**
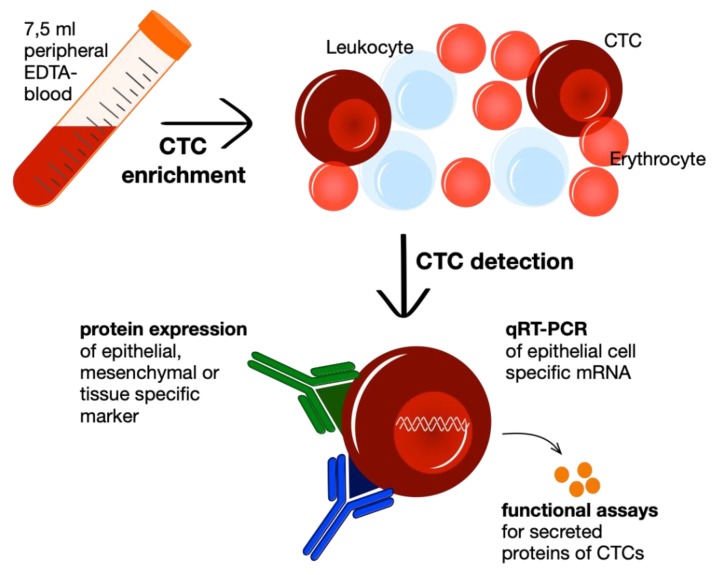
Workflow of liquid biopsy with CTC enrichment and following detection by either immunocytological detection, genome-based identification via pRT-PCR or functional secretion assays of viable CTCs.

**Table 1 diagnostics-10-00192-t001:** Clinical significance of CTCs in gastrointestinal cancers as tools for screening, staging, prognosis prediction, therapy alignment and monitoring.

	Author, Year	Entity	No. of Patients	Detection Method	CTC No. (%)	End-Point	Clinical Significance
**screening**	Kang et al., 2017 [7]	Gastric cancer	116	FAST	99 (85)	-	-
	Sefrioui et al., 2017 [8]	Pancreatic cancer	49	size-based Screencell Cyto filtration	33 (67)	-	no
	Rosenbaum et al., 2017 [9]	Pancreatic cancer	171	CellSearch	115 (67)	-	No, low specificity for PDAC (63%)
	Castro et al., 2018 [10]	Healthy people	3388	CellSearch	-	-	Ongoing ICELLATE2 study
	Yang et al., 2018 [11]	Gastric cancer	40	Microfluidic chip	20 (75)	-	yes
**staging & prognosis prediction**	Qiao et al., 2017 [12]	Esophageal squamous cell carcinoma	103	FACS	47/59 (80)	OS; PFS	yes
	Pernot et al., 2017 [13]	Gastric and esopahgeal junction cancer	60	CellSearch	-	OS; PFS; treatment monitoring	yes
	Konczalla et al., 2019 [14]	Esophageal cancer	76	CellSearch	15 (20)	OS; PFS	yes, CTC count as predictive marker in non-metastatic disease
	Reeh et al., 2015 [15]	Esophageal cancer	100	CellSearch	18 (18)	OS; PFS	yes
	Effenberger et al., 2018 [16]	Pancreatic cancer	69	CellSearch	23 (33)	OS; PFS	yes
	Buscail et al., 2019 [17]	Pancreatic cancer	22 PDAC 28 healthy controls	CellSearch/RosettSep/Oncoquick	CellSearch: Peripheral 2 (22); portal vein 5 (22)	OS; PFS	yes, combination of two sampling sites and combination with tumor exosome analysis are sensitive prognosis prediction tools
	Amantini et al., 2019 [18]	Pancreatic cancer	20	ScreenCell	20 (20)	OS; PFS; molecular expression pattern	yes
**therapy alignment &monitorig**
	Yin et al., 2012 [19]	Esophageal cancer	72	rT-PCR	-	Radiotherapy (RT) response	Yes, CTC count variation due to RT correlated with response rate
	Brabender et al., 200 [20]	Esophageal cancer	29	rT-PCR	-	-	Yes, reduced chemotherapy response in patients with ERCC1 positive CTCs
	Lankiewicz et al., 2008 [21]	Colorectal cancer	34	Multiplex PCR	20 (59)	-	Yes, CTC cound predicts chemotherapy response, moreover EGFR status of CTCs could predict likelihood of targeted therapy response
	Gazzaniga et al., 2010 [22]	Colorectal cancer	40	CELLection Dynabeads^®^	27 (68)	PFS; OS; molecular expression pattern	Yes, patients with ALDH1, survivin and MRP5 positive CTCs had significantly shorter PFS
	Takeda et al., 2019 [23]	Colorectal cancer	34	Microfluidic chips	34	-	Comparison of mutational status of CTCs, ctDNA and primary tumor tissue revealed great heterogeneity
	Iwatsuki et al., 2013 [24]	Gastric cancer	87	CellSearch	62 (71)	-	Yes, 36% of discordant HER2 status between primary tumor and CTCs, predict likelihood of targeted therapy response
	Kolodziejczyk et al., 2007 [25]	Gastric cancer	32	FACS	-	-	Yes, neo-adjuvant chemotherapy significantly reduces CTC count in responders
	Neki et al., 2013 [26]	Colorectal cancer	14	CellSearch	14; 4 (29) after chemotherapy	PFS; OS;	Yes, CTC negative patients after chemotherapy had significantly better treatment response
	Lu et al., 2011 [27]	Colorectal cancer	141	rT-PCR	141 (100)	PFS; OS	Yes, CTC persistence after surgical resection was a significant marker for early recurrence
Tol et al., 2009 [28]	Colorectal cancer	467	CellSearch	467 (100)	PFS; OS;	CTC count provides additional information to CT imaging for early recurrence monitoring

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
