# Peer review of "Clinical Significance of Circulating Tumor Cells in Gastrointestinal Carcinomas"

_diagnostics, 2020, doi:10.3390/diagnostics10040192_

Round 1

Reviewer 1 Report

I read with very interest this communication, entitled "Clinical Significance of circulating tumor cells in gastrointestinal carcinomas", and I think that is very focused on the goal of the project, with all new bibliography reported suggesting an important use of CTCs during the studies and the follow up of colon cancer disease, pre-surgery and post-surgery. I think that is possible to accept this communication in the present form.

Author Response

I read with very interest this communication, entitled "Clinical Significance of circulating tumor cells in gastrointestinal carcinomas", and I think that is very focused on the goal of the project, with all new bibliography reported suggesting an important use of CTCs during the studies and the follow up of colon cancer disease, pre-surgery and post-surgery. I think that is possible to accept this communication in the present form.

We thank reviewer 1 for this positive statement.

Reviewer 2 Report

The authors have presented a review focused on the role of CTCs in gastrointestinal cancer types. 

The topic is up-to-date as liquid biopsy techniques are currently emerging as potential useful tools also in gastrointestinal malignancies on par with other cancer types where they are used extensively even nowadays (such as in lung cancer).

Even though I like how the authors have presented the papers that have been included in the review (by focusing on different indications of CTCs monitoring in diagnostic, follow-up and therapeutic phase), I think that a few minor revisions have to be done before it can be deemed acceptable for publication.

First, there are a few areas of the manuscript where english language should be polished or some typos need to be corrected (for example Page II Line 61 "tend to be diagnosed at progressed tumor stadium" that could be better explained in "tend to be diagnosed at advanced tumor stage", and many others). English language needs some extensive revision.

Furthermore, it is not known how the authors have selected the papers to include in the review: in the authors' contribution section 3 different authors have contributed to methodology but we do not know how exactly papers to be included in the review have been selected.

There are a few interesting papers on the subject (p.e. Amantini C et al. Front Oncol. 2019 Sep 10;9:874. doi: 10.3389/fonc.2019.00874, or Buscail E et al. Cancers (Basel). 2019 Oct 26;11(11). pii: E1656. doi: 10.3390/cancers11111656., or even Takeda K. Cancer Sci. 2019 Nov;110(11):3497-3509. doi: 10.1111/cas.14186.) that have not been cited, as well as some published meta-analysis (such as in Wang Y.HPB (Oxford). 2019 Nov 27. pii: S1365-182X(19)33194-6. doi:10.1016/j.hpb.2019.11.003) that I think should at least be commented by the authors, as their results would compliment those presented by this review.

Author Response

The authors have presented a review focused on the role of CTCs in gastrointestinal cancer types. 

The topic is up-to-date as liquid biopsy techniques are currently emerging as potential useful tools also in gastrointestinal malignancies on par with other cancer types where they are used extensively even nowadays (such as in lung cancer).

Even though I like how the authors have presented the papers that have been included in the review (by focusing on different indications of CTCs monitoring in diagnostic, follow-up and therapeutic phase), I think that a few minor revisions have to be done before it can be deemed acceptable for publication.

First, there are a few areas of the manuscript where english language should be polished or some typos need to be corrected (for example Page II Line 61 "tend to be diagnosed at progressed tumor stadium" that could be better explained in "tend to be diagnosed at advanced tumor stage", and many others). English language needs some extensive revision.

            Thank you for this remark. The text has been read thoroughly again and corrected and optimized by a fluently speaking professional.

Furthermore, it is not known how the authors have selected the papers to include in the review: in the authors' contribution section 3 different authors have contributed to methodology but we do not know how exactly papers to be included in the review have been selected.

            A comprehensive search of electronic databases (PubMed, ScienceDirect, and Google scholar) using the key words “circulating tumor cells and/or liquid biopsy and/or staging and gastrointestinal cancer and/or solid cancer” and “prognosis or therapy or clinical monitoring or molecular marker” was performed. The reference lists provided by the identified articles were additionally hand-searched for additional studies missed by the search strategy, and this method of cross-referencing was continued until no further relevant publications were identified. Evidence from these data was critically analyzed and summarized to produce this article with a broad overview about the most relevant gastrointestinal tumors and CTCs. We added this paragraph to Method part at page 2-3.

There are a few interesting papers on the subject (p.e. Amantini C et al. Front Oncol. 2019 Sep 10;9:874. doi: 10.3389/fonc.2019.00874, or

Buscail E et al. Cancers (Basel). 2019 Oct 26;11(11). pii: E1656. doi: 10.3390/cancers11111656.,

or even Takeda K. Cancer Sci. 2019 Nov;110(11):3497-3509. doi: 10.1111/cas.14186.) that have not been cited,

as well as some published meta-analysis (such as in Wang Y.HPB (Oxford). 2019 Nov 27. pii: S1365-182X(19)33194-6. doi:10.1016/j.hpb.2019.11.003)

that I think should at least be commented by the authors, as their results would compliment those presented by this review.

            Thank you very much for this comment. Indeed, these papers giving great additional information, especially the study of Takeda et. al. points out the heterogeneity of primary tumor sides and CTCs. Therefore, the original articles have been included in the revised manuscript.